# Unravelling the Puzzle of Anthranoid Metabolism in Living Plant Cells Using Spectral Imaging Coupled to Mass Spectrometry

**DOI:** 10.3390/metabo11090571

**Published:** 2021-08-25

**Authors:** Quentin Chevalier, Jean-Baptiste Gallé, Nicolas Wasser, Valérie Mazan, Claire Villette, Jérôme Mutterer, Maria Mercedes Elustondo, Nicolas Girard, Mourad Elhabiri, Hubert Schaller, Andréa Hemmerlin, Catherine Vonthron-Sénécheau

**Affiliations:** 1Centre National de la Recherche Scientifique, Laboratoire d’Innovation Thérapeutique, Université de Strasbourg, CEDEX, F-67401 Illkirch, France; jb.galle@gmail.com (J.-B.G.); n.c.wasser@gmail.com (N.W.); nicolas.girard@unistra.fr (N.G.); vonthron@unistra.fr (C.V.-S.); 2Institut de Biologie Moléculaire des Plantes, Centre National de la Recherche Scientifique, Université de Strasbourg, CEDEX, F-67084 Strasbourg, France; claire.villette@ibmp-cnrs.unistra.fr (C.V.); jerome.mutterer@ibmp-cnrs.unistra.fr (J.M.); hubert.schaller@ibmp-cnrs.unistra.fr (H.S.); andrea.hemmerlin@ibmp-cnrs.unistra.fr (A.H.); 3Centre National de la Recherche Scientifique, Laboratoire d’Innovation Moléculaire et Applications, Université de Strasbourg-Université de Haute Alsace, CEDEX, F-67087 Strasbourg, France; mazan@unistra.fr (V.M.); elhabiri@unistra.fr (M.E.); 4SUPINFO École des Experts des Métiers de l’Informatique, CEDEX, F-67004 Strasbourg, France; mmelustondo@gmail.com

**Keywords:** anthranoids, antimalarial drugs, metabolism, spectral imaging, mass spectrometry

## Abstract

Vismione H (VH) is a fluorescent prenylated anthranoid produced by plants from the Hypericaceae family, with antiprotozoal activities against malaria and leishmaniosis. Little is known about its biosynthesis and metabolism in plants or its mode of action against parasites. When VH is isolated from *Psorospermum glaberrimum*, it is rapidly converted into madagascine anthrone and anthraquinone, which are characterized by markedly different fluorescent properties. To locate the fluorescence of VH in living plant cells and discriminate it from that of the other metabolites, an original strategy combining spectral imaging (SImaging), confocal microscopy, and non-targeted metabolomics using mass spectrometry, was developed. Besides VH, structurally related molecules including madagascine (Mad), emodin (Emo), quinizarin (Qui), as well as lapachol (Lap) and fraxetin (Fra) were analyzed. This strategy readily allowed a spatiotemporal characterization and discrimination of spectral fingerprints from anthranoid-derived metabolites and related complexes with cations and proteins. In addition, our study validates the ability of plant cells to metabolize VH into madagascine anthrone, anthraquinones and unexpected metabolites. These results pave the way for new hypotheses on anthranoid metabolism in plants.

## 1. Introduction

The plant kingdom is a source of ~200.000 identified specialized metabolites of which, ~10.000 are phenolic compounds also called polyphenols. Many of these metabolites are used as ingredients in the pharmaceutical, cosmetics, and agri-food industries on account of their diverse bioactive properties [1]. Anthranoids form a large class of polyphenols including anthraquinones, anthrones, and bianthrones; these are characterized by anthracene-based structures with various degree of oxidations and are conjugated with sugars and/or prenyl groups [2]. The biological action of several anthraquinones widely considered in the literature has demonstrated the need for the presence of specific functions. For instance, in the case of emodin (Emo), hydroxyl groups at position 1 and 8 of the anthracene ring (Figure 1A) are mandatory for its purgative properties [3,4]. Vismione H (VH) (Figure 1A) is a prenylated anthranoid, generating significant interest due to promising antimalarial and antileishmanial activities [5,6]. Chemical inventories of botanical resources, structural elucidation, and biological activities of natural products like VH allowed a compilation of comprehensive repositories for potential drugs. Still, fundamental questions about their biosynthesis *in planta* their molecular targets for biological activities, and their metabolization in cells into potentially active derivatives remain unsolved. In plants, anthranoids are biosynthesized through two distinct pathways: the polyketide pathway occurs in the Rhamnaceae, Fabaceae, Aloeaceae, and Polygonaceae families, while the shikimate/*o*-succinylbenzoic acid pathway is found in Rubiaceae [7]. Recently, genome mining identified new candidates for anthraquinones biosynthesis enzymes in *Senna tora* plants [8]. Anthranoid metabolism has also been reported in mushrooms belonging to the *Cortinarius* genus and in *Aspergillus nidulans* [9,10]. These data suggest that O-methylation, oxidation, hydroxylation, dimerization, and glycosylation may be enzyme-catalyzed, while other modifications could result from chemical reactions such as tautomerization, photoisomerization, and photochemical hydroxylation [11,12,13]. Under oxidative conditions such as in DMSO, the VH isolated from *Psorospermum glaberrimum* spontaneously degrades into the anthrone form, which is then oxidized into its madagascine (Mad) anthraquinone form or alternatively dimerized into bianthrones [6].

Within the entire UV-Vis. spectrum, more than 300 naturally occurring fluorescent compounds have been reported with quantum yields ranging from 0.01% to 100% in vitro [14]. In fact, the number and position of substituents, especially hydroxyl groups impact the physico-chemical (i.e., protonation) and fluorescence properties of anthranoids [14].

The acetyl vismione D emits green fluorescence (λ_Em_ = 534 nm) in methanol with a low quantum yield of about 2% [6], whereas anthrones/anthranols emit light at a blue wavelength (λ_Em_ = 458 nm) in alcohols and a strong yellow-green fluorescence in water (λ_Em_ = 539 nm) [15]. In regards to anthraquinones such as quinizarin (Qui) (Figure 1A) or hypericin, both fluoresce in the orange to far red window (570–675 nm) with quantum yields up to 30% [14,15,16,17]. Overall, the specific fluorescence properties of VH, anthrone, and anthraquinone forms have never been exploited in integrative biochemical approaches, especially to elucidate their biosynthesis, metabolism, cell compartmentation, and bioactivity in living cells.

Spectral imaging (SImaging) enables the simultaneous detection of emitted fluorescence in multiple independent channels with a resolution of <10 nm/channel (Figure 1B). Auto-fluorescent phenolic compounds such as simple phenols, vanillin, and mangiferin were indeed observed by SImaging [18,19]. Here, SImaging combined with high-resolution mass spectrometry was implemented to track VH and related anthranoids (Mad, Emo, Qui) and were compared to a prenylated naphthoquinone lapachol (Lap) in tobacco BY-2 cells (Figure 1A,B). Tobacco cells were chosen as a model because they are inexpensive, safe, easy to handle, and free of auto-fluorescent compounds under standard conditions [20]. Fraxetin (Fra) was used as a positive control because its fluorescence and metabolism are reported in Arabidopsis and tobacco BY-2 cells [21,22]. To faithfully characterize the molecules on the basis of their fluorescence observed by SImaging, we performed a spectrofluorometric analysis of the pure compounds in solution as a reference (Figure 1B). This comparison enabled the identification of natural product fluorescence signals within different cell compartments at a single time point. Combined with non-targeted metabolomics (UPLC-HRMS/MS) of treated tobacco cells, providing unique and accurate annotations of fluorescent metabolites (Figure 1C), this elegant photobiochemistry approach offers a fresh view on the fate of the antimalarial agent vismione H in living plant cells.

## 2. Results

### 2.1. PH Influences the Spectral Properties of Compounds in Solution

pH values and composition in salts differ in plant sub-cellular compartments, hinging on cell type, the developmental stage, and the environment [23,24]. As the intracellular pH in plant cells ranges from 5 in the primary vacuole, up to 8 in the mitochondrial matrix and peroxisomes [25,26], we investigated in detail the photophysical properties of anthranoids and coumarins, whose fluorescence is under physiochemical control. It is noteworthy that all studied compounds showed at least two main absorption bands, one with high absorptivity at wavelengths below 300–320 nm and another less intense band at much lower energies (from 350–600 nm, Appendix A). In all of the examined systems, a significant bathochromic shift of the absorption lying at lower energies was observed upon increasing the pH. In contrast, a hypochromic shift of the main absorption band and the emergence of a weak absorption band, centered at about 500 nm under basic conditions, was observed for VH (Appendix A). As reported for other polyphenols [27,28], these results showcase the impact of the moderate acidity and the associated hydroxyl deprotonation of the investigated polyphenols on their respective absorption spectral characteristics. In addition, the second absorption centered at 350–600 nm is appropriate to SImaging methodology settings and to those of most confocal microscopes.

Among the six compounds studied, except for Lap (Appendix A), all compounds emitted fluorescence when excited between 350–520 nm (Figure 2). In an organic solvent such as ethyl acetate, VH did not emit fluorescence as compared to anthraquinones (Appendix A), whereas Fra could only be excited by λ < 330 nm (Appendix A). Still, in saline ethanolic solution at pH 2 (i.e., fully protonated and neutral species), VH and anthraquinones were found to be fluorescent and their absolute quantum yields Φ_F_ ranged from 0.7 to 15.1% (Table 1, Figure 2A–E). Surprisingly, although Mad only differs from the Emo by the prenylated C3-OH group (Figure 1A), its absolute quantum yield Φ_F_ = 4.8% was found to be 6 times higher than that of Emo Φ_F_ = 0.7%. The 1,4-dihydroxyanthraquinone Qui was found to be the most fluorescent anthraquinone with a Φ_F_ = 15.1%. These results indicate that the number and position of hydroxyl groups as well as other types of substitutions (e.g., prenyl group on position 3 for Mad) on anthraquinones contribute to the brightness of their fluorescence emission. In addition, anthraquinones show a drastic loss of their fluorescence emission intensity (Appendix A) when the pH value is higher than the pK_a1_ (i.e., monodeprotonated species, Appendix A), and vice versa for Fra or VH (catechol for Fra or naphthalene-1,8-diol for VH, Appendix A). Nonetheless, a subsequent increase of pH to 12 and higher lead to the progressive loss of Fra and VH fluorescence (Appendix A). It can be speculated that the former likely promoted VH and Fra degradation into other compounds by lactone ring opening or anthraquinone formation from VH as previously described in DMSO [6,29,30]. Interestingly, the λ_Em_ values measured for VH in NH_4_HCO_3_ and Na_2_B_4_O_7_ buffers at pH 10 were found to be bathochromically shifted from 481 to 533 nm (LNH_4_^+^ Φ_F_ = 24.0%) and 532 nm (LB Φ_F_ = 31%), respectively (Table 1, Figure 2B and Appendix A).

In Na_2_B_4_O_7_ buffer, VH is strongly emitting as reported for anthranol in the Schouteten reaction [15]. Similarly, Fra λ_Em_ was 492 nm (LB Φ_F_ = 1.1%) and 481 nm (LNH_4_^+^ Φ_F_ = 0.8%) in NH_4_HCO_3_ (Table 1, Figure 2F and Appendix A). These results indicate that in cellular environments with acidity ranging from 4 to 7.8, all compounds except Lap and Fra would emit fluorescence if excited at λ_Ex_ ranging from 392 to 480 nm.

As far as Fra and VH are concerned, the distinct photophysical properties of the LB and LNH_4_^+^ characterized species support the fact that these two compounds are likely able to chelate boron as already reported for Mg^2+^ or Fe^2+^ with Fra and anthraquinones [21,31], but also ammonium. In this context, we carefully investigated the influence of metals (Ca^2+^ and Mg^2+^) and the BSA as model protein for chelation experiments, on the VH emission properties.

### 2.2. Metal and Protein Chelations Influence VH Fluorescence Properties

Ca^2+^ and Mg^2+^ were selected for complexation studies not only because of their abundance in plant cells, but also for their key roles in cell structure and physiology such as signaling pathways or the water splitting complex of photosystems [32,33,34]. BSA was selected as a protein model as it has already been used for protein chelation assays with other fluorescent compounds [35]. Thus, we evaluated the absorption (26 µM VH) and fluorescence (2.6 µM VH) properties of VH in EtOH/water 1:1 *v*/*v* containing 0.1 M of NaCl in the presence of Mg^2+^ or Ca^2+^. As BSA precipitated under these experimental conditions, the protein complexation studies were performed only in water. Accordingly, VH chelates both Mg^2+^ and Ca^2+^ with a comparable affinity (Figure 3, Appendix A). From the UV-Vis. absorption titrations, log *K*_VHM_ values of 2.31 ± 0.07 and 2.24 ± 0.06 respectively, were calculated and indicated that substantial amounts of VH-Ca or VH-Mg complexes can be formed within the cells. In both cases, about 70% of VH (26 µM) complexation was achieved with 6 mM of CaCl_2_ or MgCl_2_. Although VH displays similar binding strength with these metal ions, the absorption data pointed out different binding modes (Appendix A).

Marked spectral differences observed in the case of Ca^2+^ as compared to Mg^2+^ assert divergent coordination preferences. As reported, carboxylates preferentially act as bidentate binders with Ca^2+^ and as monodentate ligands with Mg^2+^ in proteins [36]. This property could explain our absorption data with VH acting as bidentate ligand with Ca^2+^ (β-hydroxy-ketone binding unit leading, Appendix A), while preferentially coordinating Mg^2+^ with monodentate binding unit mode (phenolate unit).

Fluorescence analysis of VH-Ca and VH-Mg complexes in solution confirmed the impact of chelation on photophysical properties. The maximum of emission λ_Em_ for VH-Ca and VH-Mg (549 ± 1 nm) complexes was found to be higher than that of the VH LB or LNH_4_^+^ species (532 ± 1 nm) (Table 1). Nonetheless, the Φ_F_ of VH-Ca (19.8%) and VH-Mg (18.6%) complexes were substantially lower than that of VH LB (31%) species, but still much higher than the neutral species (3.7% for VH LH_2_). We then evaluated the binding strength of VH and Fra with the BSA protein model, both by absorption and emission means (Appendix A). VH and Fra were found to strongly interact with BSA, with stability constants log *K*_VHBSA_ and log *K*_FraBSA_ values of 5.3 ± 0.2 and 5.7 ± 0.3 (Appendix A), respectively. This suggests that VH and Fra would interact mainly with proteins rather than with divalent metal ions in cellula. VH chelates BSA protein with a weak alteration of the absorption properties as seen previously with Mg^2+^ (Figure 3B and Appendix A), while the absorption properties of Fra were significantly altered (Appendix A). The λ_Em_ of VH LBSA fluorescence spectrum corresponded to 517 nm with Φ_F_ = 23.5%, close to that of VH LNH_4_^+^ species (Table 1, Appendix A), while Fra-BSA result was 10 times higher than that of Fra LNH_4_^+^ species (Table 1, Appendix A). It can be proposed that electrostatic interactions with ammonium residues such as those found in lysine or arginine, allow complexation of VH with proteins as compared to Fra which seems to interact with other amino acid residues. In contrast, Emo and Qui emissions were almost unaffected in the presence of BSA (Appendix A), as reported for complexes with DNA [17]. Overall, although neutral VH and Fra LH_2_ with low Φ_F_ seemingly predominate at intracellular pH, our results suggest that interactions with endogenous metal ions or proteins might improve/modulate their fluorescence properties in cellula.

### 2.3. Fluorescent Anthranoids Metabolization and Transport in Cellula by Spectral Imaging

To characterize the fluorescence in cellula, we compared the reference emission spectra obtained from our spectrofluorimetric analysis to those measured by SImaging in solution and in living BY-2 cells. SImaging methods were used to discriminate mixed fluorescent signals from the studied compounds in solution and in BY-2 cells with a special focus on vismione H (blue to green-yellowish emission) and related anthraquinones (yellow to red emission) (Figure 1A). With respect to the photophysical data, two different settings were selected for the excitation and detection of fluorescence using. SImaging: λ_Ex_ = 405 nm (λ_405_) with emission spectra ranges from 415 to 664 nm, and λ_Ex_ = 488 nm (λ_488_) with emission spectra ranges from 495 to 664 nm. The fluorescence data obtained by SImaging on pure compounds in solution at λ_405_ and λ_488_ fit well with those obtained by the spectrofluorimetric approach (Figure 2, Figure 3 and Appendix A). However, some variations can be observed such as a decrease in the intensity of the shoulder of the Emo LH_3_ at 575 nm or a shift from 484 to 503 nm for the VH LH_2_ or from 492 to 521 nm for the Fra LB. These discrepancies could be related to differences in resolution, glass support or optical path between SImaging (9 nm, glass microscope slide, 1 mm) and spectrofluorimetry (1 nm, quartz cell, 1 cm). In addition, the hydrophobic character of Mad was appreciated by the observation of aggregates at 1 mM Mad in hydro-alcoholic solution at pH 2 (Appendix A). Although the Φ_F_ of Mad is higher than that of Emo, its lower solubility results in a lower fluorescence signal, which reduces the quality of spectra recorded at λ_488_ or λ_405_ for Mad in solution. To note, this solubility problem was also observed in the cell culture medium after treatments, leading to the formation of aggregates still present after 18 h.

In vivo, very low fluorescence was detected in control (Ctr) and Lap-treated cells (Figure 4F, Appendix A). We defined these signals as the autofluorescence threshold in tobacco BY-2 cells. For all other compounds, the emergence of fluorescence was clearly observed in treated BY-2 cells (5 min) being stable after 18 h treatments with either 25 µM (Figure 4A–E) or 50 µM (Appendix A). Overall, excitation at λ_405_ resulted in a stronger fluorescence signal than at λ_488_, but the opposite effect was observed with Qui- (Figure 4B) and Mad-treated cells (Figure 4D). As evidenced by our spectral analysis, only Fra-treated cells did not display fluorescence at a λ_488_ excitation (Figure 4E). Except for VH, all spectra recorded at 5 min after treatment were very similar (same λ_Em_ and shape) and independent of λ_Ex_ (Figure 4B–E and Appendix A). PCA of the normalized average spectra support the idea that the emission spectrum observed at λ_488_ in VH-treated cells shares similarities with those measured with Qui and especially Mad, but not with spectra found in Emo-treated cells (Figure 5B and Appendix A). Besides, the emission maxima observed in VH and Mad-treated cells (mostly in vesicular bodies) following excitation at λ_488_, were centered at 539 nm and 575 nm. Even though the emission spectra recorded for VH- and Mad-cells share similarities with that of Qui-treated cells, the maximum of emission for Qui-treated cells was centered at 575 nm. These results strongly suggest that within the first minutes, VH and Mad lead to similar anthraquinones differing from those produced in both Emo- and Qui-treated cells. Interestingly, the fluorescence emission detected was usually observed first in the cytoplasm (Appendix A), and after 5 min in additional structures such as Golgi bodies (Appendix A), lipid droplets (Appendix A), and the ER (Appendix A). Specifically for Fra-treated cells excited at λ_405_, a fluorescence emission was observed in the nucleus. Therefore, molecules are well absorbed by BY-2 cells and diffuse in different cell compartments following their polarity. The difference of shape and intensity between spectra collected in VH-treated cells suggests that at λ_405_ the monodeprotonated or complexed VH is detected in the cytoplasm and the ER, while at λ_488_ anthraquinones are predominantly observed in vesicular bodies such as lipid droplets and some Golgi bodies. After 18 h, at λ_488_ or λ_405_ slight variations were detected in fluorescence intensities.

In particular for anthranoids, it was found that their localization and emission spectra did not change significantly over time. Nevertheless, at λ_405_, the shape of the emission spectrum in Mad-treated cells was found to be closely related to that of VH (Figure 4A,D). Other treatments have also been associated either with the appearance of a new fluorescence emission signal in the primary vacuole (Figure 4A–C and Appendix A), or to the translocation of an identical fluorescence signal from the nucleus into the primary vacuole as seen in Fra-treated cells (Figure 4E and Appendix A). These results were further validated by a PCA analysis of the normalized average spectra (Figure 5C,D, Appendix A), giving rise to clustering of fluorescence spectra according to putative structural similarities between the tested molecules in BY-2 cells. For instance, at λ_488_ (Figure 5D), the normalized average spectra of anthranoid-treated cells clustered, indicating that anthraquinones fluorescence was observed in contrast to Ctr, Lap, and Fra, which do not fluoresce under these conditions. In addition, normalized average spectra collected at λ_405_ (Figure 5C) and at λ_488_ (Figure 5D) within the cytoplasm, ER and vesicular bodies in VH, and Mad-treated cells were found to be similar and even closer after 18 h (Figure 5C,D) than after 5 min (Figure 5A,B). Therefore, they clustered in the PCA analysis. Conversely, the Emo normalized average spectra are significantly different at λ_405_ (Figure 5C) and slightly less different at λ_488_ (Figure 5D)_._

Taken together, these results strongly support that VH was transformed in vivo into anthraquinones with a structure closer to that of Mad than to that of Emo. In addition, PCA of standardized average spectra with excitation at 405 nm (λ_405_) clearly confirmed the appearance of a new signal observed in the primary vacuole after 18 h treatment with VH-, Emo-, and Qui-treated cells (Figure 5C), whereas it was not observed using excitation at 488 nm (λ_488_).

The fluorescence detected in the primary vacuole is similar to that detected in the cytoplasm for Emo and Qui unlike VH, for which the fluorescence observed in the primary vacuole is significantly different from that measured in the cytoplasm, ER and vesicular bodies. As a partial conclusion, SImaging analyses allowed us to accurately track the fluorescence of VH, related anthranoids and the coumarin (i.e., fraxetin Fra) compared to Lap which exhibits no fluorescence in living cells. Moreover, the new fluorescence spectra observed after 5 min and 18 h supports the hypothesis that VH is metabolized into Mad anthraquinone-types in vivo. Interestingly, we detected signals at λ_405_ located in cell compartments whose acidity is below p*K*_a1_ values in Fra- and VH-treated cells (i.e., p*K*_a1_ = 8.5 for Fra and pK_a__1_ = 7.2 for VH). According to fluorescence of Fra and VH in model solution (Table 1), only complexes and especially LBSA species emit strong fluorescence at neutral pH. Thus, the bright signal observed at λ_405_ after feeding with VH and Fra strongly support that part of Fra and VH are complexed in cellula and/or metabolized into related fluorescent compounds. 

### 2.4. Fra and Methyl-Fra Derivatives Prevail to Fra-Glycosylated Forms

An increase in the polarity of given compounds typically results from their oxidation by oxygenases or glycosylation by glycosyltransferases. This metabolization enables then the sequestration of phenolic compounds into the primary cellular vacuole as described with Fra [22,37]. To identify metabolites of Fra, Qui, Emo, Mad, and VH characterized by the fluorescence spectra detected by SImaging, non-targeted metabolomics of methanolic extracts from treated and non-treated BY-2 cells was carried out using UPLC-HRMS/MS. The results were compared to a database including the reference compounds and related metabolites deduced from in silico biotransformation. In this regard, 54 metabolites absent in the control extracts were annotated according to *m/z* of the parent ion and isotopic profile as compared to references or putative catabolites. In addition, hypothetical isomers and/or conjugates were annotated according to MS/MS fragments. As it cannot be excluded that after separation a loss of conjugates occurred in the MS source, the metabolites with different retention times (R_t_) but similar *m/z* and MS/MS fragments were annotated as derivatives represented by putative isomers and/or conjugates. 

All references except Lap (Appendix A) were identified in corresponding methanolic extracts of BY-2 cells treated for 15 min or 18 h (Figure 6 and Appendix A). The coumarin mixture annotated in the methanolic extracts from Fra-treated cells (Figure 6A) consists of 12 tri-oxygenated forms distributed in Fra and its supposed isomers (F1–F3), six methylated (F4–F9) and three glycosylated forms (F14–F16). The remaining metabolites correspond to tetra-oxygenated coumarins including reduced sideretin and two derivatives (F10–F12) together with another methoxylated form (F13). After 15 min, Fra quickly forms the more polar F2 derivative. Simultaneously, both compounds may have been methylated (F8, F9) or glycosylated (F15, F16).

After 18 h, Fra methylated forms F8 and F9 were significantly reduced in contrast to F2 derivative and F15 or F16 glycosylated forms, remaining constant (Figure 6A). In addition, the Fra derivative F1 abundance was increased 7-fold after 18 h, and other new methylated (F4, F5), hydroxylated (F10), as well as glycosylated (F14) forms were 6 to 33- fold more abundant (Figure 6A and Appendix A). Thus, the annotated metabolites are consistent with those reported for BY-2 cells treated for 60 min with 20 µM of Fra [22]. Although the absolute quantitation was not performed, the Fra isomers or conjugates appeared to be a prevalent form for storage in the primary vacuole, while abundance of glycosylated derivatives did not significantly change over time. In comparison, Lefèvre et al. [22] reported that 7 days-old BY-2 cells treated with Fra (20 µM) accumulate 63% of glycosylated derivatives. The metabolomics and SImaging results suggest that the spectral fingerprint observed at λ_405_ after 5 min and 18 h Fra treatments corresponds to a mixture of more polar Fra related metabolites rather than Fra itself. In this respect, Fra is metabolized and translocated from the cytoplasm and the nucleus to the primary vacuole for storage/sequestration in BY-2 cells (Figure 4E and Figure 6A,D). However, it was noticed that each group is mostly represented by a metabolite with an intermediate R_t_ (F2, F6, F11, F15) (Figure 6A, Appendix A). Although BY-2 cells tend to produce other forms with lower R_t_ (more polar), it can be speculated that the latter cannot be over accumulated due to putative cytotoxic effects. This is supported by the accumulation of methylated forms after 18 h, the latter being also reported to decrease negative effects of free hydroxyls from 1-hydroxycantin-6-one in *Ailanthus altissima* [38].

### 2.5. Metabolization of VH into Anthrones and Anthraquinones

The metabolomic analyses of extracts isolated from anthranoid treated cells after 15 min and 18 h highlighted specificities. For instance, Qui (Q6) and five of its derivatives (Q1–Q5) were found exclusively in Qui-treated cell extracts (Figure 6B, Appendix A) while Emo-, Mad-, and VH-treated cells shared a few common metabolites (Figure 6C, Appendix A). We noted an absence of methylated, hydroxylated or glycosylated forms, which may be explained by the unusual 4-OH group of Qui found in traces as reported in *Cassia obtusifolia* extracts [16]. In contrast, most natural anthranoids are found to be hydroxylated at all other positions except the position 2 associated to carboxylic acid progenitors [9]. Accordingly, it can be speculated that Q1–Q5 are five isomers resulting either from a chemical tautomerization/isomerization or from redox processes [11,39]. Meanwhile, it cannot be excluded that 1-OH was removed and the position C5 or C7 hydroxylated as reported for Emo [9,10]. The specific detection of Qui in related methanolic cell extracts is consistent with its fluorescence detected by SImaging at λ_488_. However, the non-targeted metabolomics pointed out that the Qui fluorescence detected in the cytoplasm, lipid droplets and some Golgi bodies may be also related to Q1–Q5 derivatives. In the absence of Q1–Q5 references, we were not able to confirm whether these derivatives or Qui contributed to the fluorescence observed and especially that in the primary vacuole (λ_405_). Thus, it can also be assumed that the latter would have originated from other non-annotated metabolites.

In VH-, Mad-, Emo-treated cells, the Emo (A8) and eight of its derivatives (A1–A7, A9) were detected (Figure 6C, Appendix A). In addition, five methylated (A10–A14), two hydroxylated (A15, A16), two methoxylated (A17, A18), and four glycosylated (A24–A27) Emo derivatives were significantly detected only in Emo-treated cells. Surprisingly, Emo and its A1–A6 derivatives were almost exclusive of Emo-treated cells and after 18 h the A1–A5 abundance significantly increased. In contrast, the Emo derivatives A7 and A9 were specific to Mad and VH-treated cells. This agrees with SImaging and PCA results obtained for Emo-treated cells at 5 min and 18 h. Indeed, the fluorescence observed mostly in the cytoplasm at λ_405_ and λ_488_ corresponds to a specific spectral fingerprint of Emo and more likely to A6 content that is unchanged in our metabolomic data at both 15 min and 18 h. In contrast, it can be proposed that the fluorescence observed in the primary vacuole at λ_405_ after 18 h, is from a mixture of the most hydrophilic derivatives (A1–A5), glycosylated (A24, A25) or hydroxylated (A15, A16) forms, being significantly increased after 18 h as compared to 15 min (Figure 6C,D). Other metabolites such as VH (A23) as well as Mad anthrone and two putative isomers (A19–A21) were only detected in VH-treated cells, while Mad (A22) was found in VH and Mad-treated cells.

## 3. Discussion

For years, the metabolism, biosynthesis and bioactivity of anthranoids were studied through chemical analysis of metabolites after feeding experiments with radiolabeled precursors [9], genome mining [8,10], and bioassays [5,6], respectively. Our study demonstrates that SImaging is an additional sensitive and suitable tool for the observation of anthranoids in vivo. In depth, unlike classical microscopy, SImaging allowed us to localize and discriminate VH and related fluorescent metabolites over time, in living cells. Still, UV-Vis spectrophotometric and spectrofluorometric analyses were crucial not only to characterize unbiasedly the fluorescence of compounds in solution, but also to evidence the influence of pH or complex formation with cations and protein on it. In fact, SImaging had already been used to study the role of coumarins such as scopoline, fraxin and esculin in the iron metabolism of *A. thaliana* roots [40], but not fraxetin Fra (the fraxin genuine). Even though Fra is reported as a non-fluorescent coumarin [21,22], our study evidenced that under alkaline or complexation conditions, its fluorescence is enhanced and modulated depending on the ligand (Table 1). Surprisingly, the Fra fluorescence spectra recorded by SImaging and spectrofluorimetry were significantly different except for Fra-BSA complex (Table 1, Figure 2F, Appendix A). In this context, the overlapping of fluorescence spectra from free and complexed coumarins is limiting for discrimination by SImaging. In fact, extracting spectra to unmix fluorescence signals has been reported (Robe et al., 2021), however our data indicate a cellular colocalization of molecular species that are therefore barely distinguishable even with acquisition settings of the fluorescent signal at 9 nm/channel. By contrast, the anthranoid fluorescence spectra are consistent between SImaging and spectrofluorimetry analyses, allowing good identification of the fluorescent species observed. The fruitful combination of SImaging with non-targeted metabolomics was particularly effective in identifying at 5 min and after 18 h, not only key anthranoid metabolites such as madagascine anthrone, Mad and putative emodin derivatives, but also a battery of methylated, hydroxylated, and glycosylated Fra-derivatives. To note, except sideretin, the methylated and hydroxylated Fra-derivatives annotated in our analysis, were not taken into account in the study describing coumarin/iron metabolism in *A. thaliana* [21,40]. As coumarin accumulation and trafficking is a complex and dynamic process, these derivatives also previously reported in *N. tabacum* [22], might be considered for further study in *A. thaliana* as well. Therefore, non-targeted metabolomics shows a clear advantage in elucidating unraveled metabolic pathways and identifying unintended and scavenged metabolites prior to targeted metabolomics, being more adapted for systematic quantitation of known metabolites.

For the first time, our study provides an overview of both metabolization processes and subcellular compartments implied in anthranoid metabolism in plants, involving at least five cell compartments after VH treatments (primary vacuole, cytoplasm, ER, Golgi bodies, and lipid droplet vesicular bodies). The cell compartments labelled by anthranoids are consistent with the ER, the cytosol, and the primary vacuole suggested by Han et al. in Rubiaceae [7], and the phenotype observed in melanoma cell culture treated with 10 µM Emo or Qui [17]. Collocation experiments with plastids marker was not performed, but in addition to Golgi bodies and lipid droplets, a part of the spherical structures labelled by anthranoids could correspond to plastids as reported in *Morinda citrifiolia* [41]. Outstandingly, the acetyl vismione D (C3-O-geranyl) was assumed to be not fluorescent enough for observation with fluorescence microscopy [6], here, we characterized the fluorescence in cellula of VH as well as its degradation products known as madagascine anthrone and anthraquinone. VH is not only converted to Mad anthrone and Mad under oxidative conditions such as in DMSO indeed, but in vivo too. Interestingly, the anthrone A20 remains well detected, whereas in DMSO Mad and bianthrone prevail after 8 h [6]. After 18 h the amount of VH, Mad, and A20 remained insignificantly altered in VH-treated cells, while the Mad anthrone derivatives A19, A21 were not detected anymore and Mad abundance considerably decreased in Mad-treated cells after 18 h. Unexpectedly, Mad, which is the less polar molecule (R_t_ = 11.73 min) was metabolized into the A9 derivative rather than Emo (A8), suggesting that in BY-2 cells other reactions occur with Mad prior to loss of the prenyl moiety. In this respect, the bright fluorescence in VH-treated cells observed by SImaging at λ_405_ mainly in the cytoplasm (Appendix A), Golgi bodies (Appendix A), and the ER (Appendix A) corresponds to a mixture of anthranol forms of Mad anthrone and VH and/or complexes, the anthrones being weakly fluorescent [15]. It can be proposed that pH, intracellular O_2_, redox levels, or chelation with cations or proteins may stabilize these forms [15,29,30], detected by SImaging in the different subcellular compartments. However, we could not discriminate if the fluorescence observed at λ_405_ in the primary vacuole of VH-treated cells after 18 h corresponded to a non-annotated metabolite or to VH-LH_2_ and/or Mad anthrone neutral forms. Since the metabolomic analysis evidenced similar anthraquinones between Mad and VH-treated cell extracts, the spectral fingerprint observed by SImaging at λ_488_ corresponds to Mad and likely the specific Emo A9 and/or A7 derivatives.

Taken together, these results point out that C3-OH substitution of anthranoids is a key position influencing both the anthranoid fluorescence and metabolization in BY-2 cells depending on its nature. A free 3-OH allows anthranoid metabolization into very polar Emo derivatives (A1–A6) as well as hydroxylated, methylated, or glycosylated forms dedicated to a vacuolar storage. In contrast when this position is substituted with a prenyl group as for VH or Mad, the metabolization is markedly altered producing hydrophobic Emo derivatives (A7 and A9). As we were not able to decipher whether enzymatic or chemical reactions contributed to putative oxidation and/or isomerization processes, we can only pinpoint that C3-O-prenylated anthranoid metabolization calls for targeting to lipid droplets, Golgi bodies, and to the ER subcellular location. On a recurring basis, because VH and Mad anthrone were automatically detected in VH reference solution, it can be speculated that both chemical isomerization/tautomerization and VH deacetylation occur in vivo too. However, the remarkable distribution of methylated, hydroxylated, glycosylated, and other derivatives in most treatments is consistent with a well-orchestrated subset of enzymes as it has been described for Fra or Emo [9,21]. To clarify these aspects, in the future it would be interesting to purify fluorescent labelled organellar fractions and analyze the transcriptome and proteome. From a general point of view, the total contents of metabolites detected in Emo and Qui-treated cells were doubled after 18 h, halved for Mad- or VH-treated cells and remained constant in Fra-treated cells (Figure 6D). Although, related metabolites seem to be accumulated over time in Emo, Qui, and Fra-treated cells, the abundance of references was 26, 3.5, and 16 times lower after 18 h, confirming their metabolization/storage. In contrast, the Mad and VH abundance in treated cells changed weakly, supporting the fact that both VH and Mad are transformed in vivo into non-annotated metabolites and the metabolization rate of prenylated anthranoids in BY-2 cells is limited.

Interestingly, our data unambiguously evidenced that tobacco BY-2 cells are competent to take up and metabolize anthranoids, not produced by *N. tabacum* plants. Due to its low cost, low autofluorescence, and heterogeneous metabolism, the BY-2 cell model is of particular interest and can undoubtedly be used to study metabolism and localization of other molecules of interest in living organisms. As long as anthranoids and other naturally occurring fluorescent compounds are valuable, it would be interesting to extend this study to investigate the unknown aspects of the biosynthesis pathways and the influence of biotic and abiotic factors on the metabolism of anthranoids and other fluorescent compounds. From another point of view, the developed approach offers interesting perspectives in the medical field, as VH and similar compounds have been described for their antimalarial properties [2,5,6], but the biological targets as well as mechanisms of action remain unknown. Finally, it would be interesting to reproduce experiments in red blood cells infected by *P. falciparum* treated with VH and analogs, therefore opening an avenue to fluorescence localization/structure/activity relationships studies. To be deciphered, the complexity of metabolic pathways requires more than ever multifactorial analyses of living organisms, this SImaging approach coupled with non-targeted metabolomics allows efficient characterization of subcellular location and bioconversion of fluorescent metabolites in living plant cells.

## 4. Materials and Methods

### 4.1. Chemicals

Fraxetin (7,8-dihydroxy-6-methoxy-coumarin), quinizarin (1,4-dihydroxy-anthraquinone), emodin (1,3,8-trihydroxy-6-methyl-9,10-anthracenedione), lapachol (2-hydroxy-3-(3-methyl-2-butenyl)-1,4-naphthoquinone) were purchased from Sigma-Aldrich. Vismione H and madagascine were obtained from PGE2 fraction of *Psorospermum glaberimum* as previously described [6]. Other chemicals were from the usual commercial sources with the highest purity available.

### 4.2. Spectrofluorimetric Analyses

First, the protonation properties (pK_a_ values) of the selected compounds were measured in a solvent made of 50% of EtOH and 50% water (by volume, Appendix A). The latter solvent was used for solubility reasons of the selected compounds. Then, different solutions of the pure compounds at 0.01 mg/mL were prepared from stock solutions in EtOH at 1 mg/mL and then diluted either in 2 mL of EtOH/H_2_O 1:1 v/v, 0.1 M NaCl adjusted with HCl 10^−2^ M (pH 2), Na_2_B_4_O_7_ 0.01 M or NH_4_HCO_3_ (pH 10), NaOH 10^−2^ M (pH 12) or in EtOAc. The absorbance spectra of (de)protonated species in solution were measured from 260–800 nm using a Cary 5000 UV-Vis.-NIR spectrophotometer (Agilent, Santa Clara, CA, USA) prior to any fluorescence analysis. Fluorescence spectra of solutions diluted 10-fold were recorded with 3.5 mL Suprasil^®^ quartz optical cells of 10 mm pathlength using a LS-50B spectrofluorimeter monitored with UVWinlab 5.1 software (Perkin Elmer, Waltham, MA, USA). For each compound, the fluorescence emission spectrum was recorded by exciting close to or at the maximum absorption wavelength. The instrumental parameters were adjusted to a scanning speed of 400 nm/min and excitation/emission bandwidths adjusted between 4.5 and 15 nm depending on the conditions. The fluorescence spectra of each compound were established by successive determination of the excitation (λ_Ex_) and emission (λ_Em_) maxima. A FluoroMax-4 spectrofluorimeter (HORIBA, Kyoto Japan) was then employed to determine as accurately as possible the quantum yields (Φ_F_) of anthranoids at 0.001 mg/mL with the exception of Emo (0.0015 mg/mL) in solutions at pH 2, then at pH 10 only for VH and Fra. The Φ_F_ values of the fluorescent species were calculated by using the equation below with either rhodamine 6G (R6G) or cresyl violet references.
Φ_F_ = Φ_R_ (∫ (I_F_ × A_R_ × N_F_^2^)/∫ (I_R_ × A_F_ × N_R_^2^))

Φ_R_ corresponds to the quantum yield of reference. The indices I_F_ and I_R_ denote sample and reference, respectively. The integrals over I represent areas of the corrected emission spectra, A_R_ and A_F_ are the optical density at the excitation wavelength for reference and sample, N_R_ and N_F_ correspond to the refractive index of the reference and the sample solutions, respectively.

### 4.3. Ca^2+^ and Mg^2+^ Chelating Assay

Fresh stock solution of VH (2.6 mM) in EtOH was further diluted 100-fold in 2 mL of EtOH/H_2_O 1:1 *v*/*v* containing 0.1 M NaCl. UV-Vis spectrophotometric titrations of the solutions were then carried out by adding increasing amounts of CaCl_2_ or MgCl_2_ and monitored using a Cary 5000 UV-Vis-NIR (Agilent, Santa Clara, MA, USA). Then, 0.1 M CaCl_2_ or MgCl_2_ (25 µL) prepared in water was successively added to 2 mL of the ligand solution (VH: 26 µM). The Ca^2+^ and Mg^2+^ chelating properties of VH were also investigated by fluorescence emission, by adding 150 µL of 0.1 M CaCl_2_ or 300 µL of 0.1 M MgCl_2_ solutions to a 2.6 µM VH solution. The Φ_F_ values of the Ca^2+^ and Mg^2+^ chelates with VH were measured as described in the spectrofluorimetric analysis section.

### 4.4. Plant Material and Treatment

The *Nicotiana tabacum* cv. Bright Yellow 2 (tobacco BY-2) cell suspension was made available by Toshiyuki Nagata (Tokyo University, Tokyo, Japan) and cultivated at 26 °C, on a rotary shaker set at 140 rpm in the dark, in modified Murashige and Skoog (MS) medium as reported [42]. For treatments, 7-day old cells were diluted five-fold into fresh MS medium and distributed (3 mL) in 6-well culture plates (Sarstedt, Nümbrecht Germany) containing 25 or 50 µM of VH, Mad, Emo, Qui, Fra or Lap. SImaging acquisitions were carried out either after 5 min incubation or after 18 h.

### 4.5. Spectral Imaging (Simaging) Microscopy

Treated cells or pure compounds at 1 mM in the solutions at pH 2, pH 10, and pH 12 were observed using a LSM780 confocal laser microscope (Carl Zeiss, Jena, Germany) equipped with an inverted Zeiss AxioObserver Z1, a Plan-Apochromat 20x/0.8 M27 objective, a numerical zoom adjusted to 2.8 with a laser strength of 5%. Images and emission spectra were acquired using the excitation wavelengths at 405 (λ_405_) and 488 nm (λ_488_) with the emission light collected into multiple channels incremented by 9 nm from 415 to 664 nm and 498 to 664 nm, respectively. The lambda view images correspond to superimposed fluorescence recorded in each channel according to the natural light spectrum. The spectral analysis was performed after the extraction of emission spectra from images by manual component extraction of 1 µm circles in different cell compartments labelled by a fluorescence. Images were exported from Zen v2 software (Zeiss, Jena, Germany) and assembled in the figure using ImageJ v1.53d.

### 4.6. Spectral Data Analysis

The dataset consisted of 10 spectra/cell collected from five cells in three independent biological replicates for each treatment, at 25 and 50 µM, after 5 min and 18 h. Intensities per channels of each spectrum were averaged per cell and normalized to 1 before statistical analysis with R software V4.0.0 (GNU GPLv2, R Core Team) and RStudio V 1.2.5001 (AGPLv3, RStudio Team (2020). RStudio: Integrated Development for R. RStudio, PBC, Boston, MA, USA http://www.rstudio.com/) using the ChemoSpec package V5.2.12. A distant matrix was established for each dataset applying the Pearson’s correlation coefficient, and a robust principal component analysis (PCA) was performed. The same procedure was used with solutions of pure compounds to analyze 10 spectra/acquisition in triplicate.

### 4.7. Non-Targeted Metabolomic Analysis

Freeze-dried BY-2 cells (25 mg) treated for 15 min or 18 h with 50 µM of VH, Mad, Emo, Qui, Fra, or Lap were extracted three times in 300 µL MeOH, each was sonicated during 20 min at 80 kHz (FisherbrandTM S-series) and then the extracts were filtered prior to analysis. Standard solution at 0.002 mg/mL in EtOH and MeOH extracts from three independent biological replicates were analyzed by the non-targeted metabolomics approach performed on the UltiMate 3000 UHPLC system (Thermo, Waltham, MA, USA) coupled to the ImpactII (Bruker, Billerica, MA, USA) high resolution Quadrupole Time-of-Flight (QTOF) as previously described [43]. Samples were kept at 4 °C, 3 µL was injected with a washing step after sample injection with a wash solution (H_2_O/MeOH, 90/10, *v*/*v*, 150 µL). The spectrometer was equipped with an electrospray ionization (ESI) source and operated in positive ion mode on a mass range from 20 to 1000 Da with a spectra rate of 2 Hz in AutoMS/MS fragmentation mode. The end plate offset was set at 500 V, capillary voltage at 2500 V, nebulizer at 2 Bar, dry gas at 8 L.min-1 and dry temperature at 200 °C. The transfer time was set at 20–70 µs and MS/MS collision energy at 80–120%. The MS/MS cycle time was set to 3 s, absolute threshold to 816 cts and active exclusion was used with an exclusion threshold at 3 spectra, release after 1 min and precursor ion was reconsidered if the ratio current intensity/previous intensity was higher than 5. Raw data were processed in MetaboScape 4.0 software (Bruker): molecular features were considered and grouped into buckets containing one or several adducts and isotopes from the detected ions with their retention time and MS/MS information when available. The parameters used for bucketing are a minimum intensity threshold of 1000, a minimum peak length of 3 spectra, a signal-to-noise ratio (S/N) of 3 and a correlation coefficient threshold set at 0.8. The [M+H]^+^, [M+Na]^+^, [M+K]^+^, and [M+NH_4_]^+^ ions were authorized as possible primary ions. The obtained list of buckets was annotated using a custom analyte list derived from in silico predicted metabolites (catabolites and conjugates) of the compounds of interest. The in silico prediction was performed using MetabolitePredict 2.0 (Bruker) as previously described [44]. Briefly, 79 biotransformation rules were used to predict metabolites over 2 generations. The maximum allowed variation on the mass (*m/z*) was set to 3 ppm, and the maximum mSigma value (assessing the good fitting of isotopic patterns) was set to 30. The changes in abundance of each metabolite annotated were determined using statistical analysis by comparing the area obtained under the chromatogram curve of the different metabolites analyzed in triplicates. The homogeneity of the variance was checked with a Levene test prior to a Kruskal–Wallis test, followed by multi comparison procedure using post-hoc Dunnett’s test.

## Figures and Tables

**Figure 1 metabolites-11-00571-f001:**
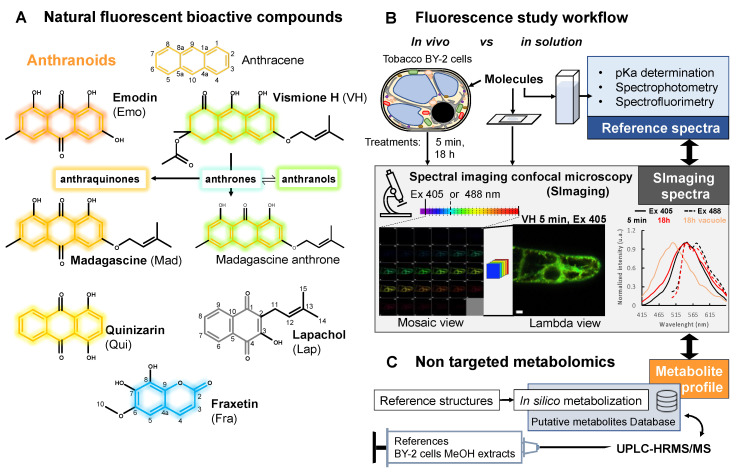
Overview of natural fluorescent bioactive anthranoids and the original approach used to study their localization and biotransformation in living cells. (**A**) Chemical structures of anthranoids characterized by the anthracene skeleton (orange skeleton) as compared to other phenolic compounds such as the naphthoquinone lapachol (Lap, grey skeleton) and the coumarin fraxetin (Fra, blue skeleton). In dimethylsulfoxide (DMSO), vismione H (VH) degrades quickly into anthrones (weak turquoise fluorescence) being in equilibrium with anthranol tautomers (strong green yellowish) and oxidized into anthraquinones such as emodin (Emo), madagascine (Mad) or quinizarin (Qui) (yellow to red fluorescence). (**B**,**C**) Spectral imaging and non-targeted metabolomic workflow to characterize biotransformation of fluorescent anthranoids in tobacco BY-2 cells. (**B**) Fluorescence of selected anthranoids is evaluated in solution, then these anthranoids (25 or 50 µM) are used to feed cells for confocal spectral imaging microscopy observations at 5 min and 18 h after feeding. (**C**) Non-targeted metabolomics workflow implementing high-resolution mass spectrometry analysis of methanolic extracts from cells characterized in (**B**).

**Figure 2 metabolites-11-00571-f002:**
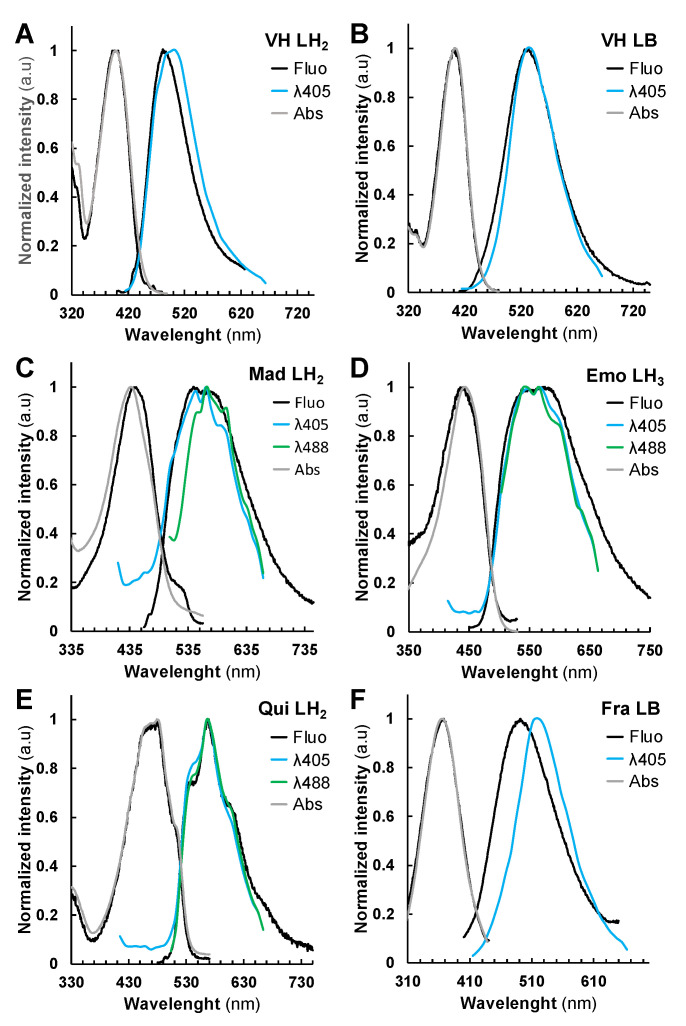
Normalized fluorescence excitation/emission and absorption spectra of studied compounds. (**A**) vismione H VH (neutral LH_2_ species) and (**B**) (boron complex LB species), (**C**) quinizarin Qui (neutral LH_2_ species), (**D**) emodin Emo (neutral LH_3_ species), (**E**) madagascine Mad (neutral LH_2_ species), (**F**) fraxetin Fra (boron complex LB species) in saline ethanolic solutions. Excitation and emission spectra obtained from the spectrofluorimetric analysis (black), absorption spectra obtained from the UV-Vis. analysis (grey), and SImaging at λ_405_ (blue) and/or λ_488_ (green) settings.

**Figure 3 metabolites-11-00571-f003:**
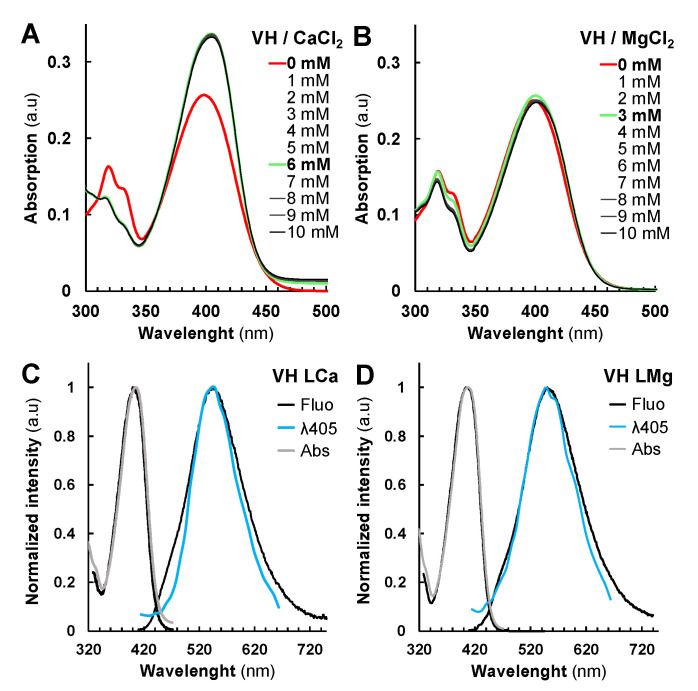
Vismione H VH photophysical properties are influenced by metal cations complexation. (**A**,**B**) Absorption spectra of metal complexes formed by VH with different amount of Ca (II) (**A**) and (**B**) Mg (II) spectra with no complex formed (bold red) and with the highest changes in the absorption spectrum (bold green). Normalized fluorescence (black), absorption (grey) and SImaging spectra of VH-Ca (**C**) and (**D**) VH-Mg complexes at λ_405_ settings.

**Figure 4 metabolites-11-00571-f004:**
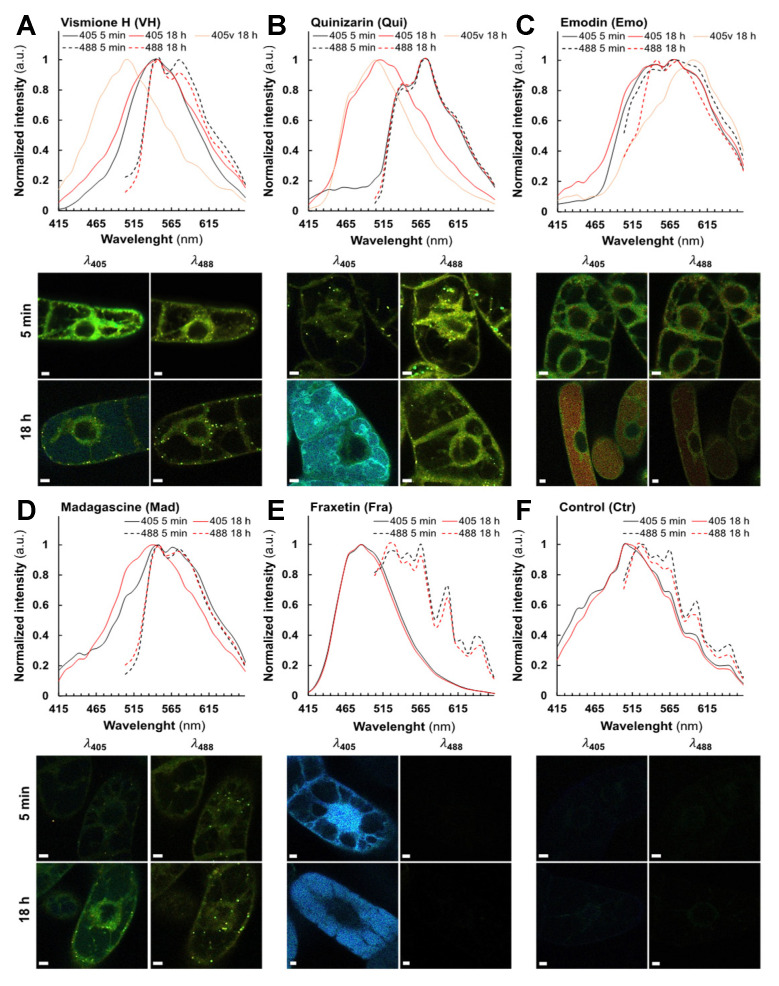
Fluorescence spectra and images of BY-2 cells treated by 25 µM of phenolic compounds prior to observation using SImaging. Normalized fluorescence average spectra and lambda view images from SImaging analysis at λ_405_ (solid line) and λ_488_ (dashed line) of BY-2 cells treated for 5 min and 18 h with 25 µM of (**A**) vismione H, (**B**) quinizarin, (**C**) emodin, (**D**) madagascine, (**E**) fraxetin, and (**F**) the negative control without treatment. Spectra observed after 5 min (black), after 18 h (red) and in the vacuole after 18 h (red light). Spectra observed in the primary vacuole at λ_405_ after 18 h are specified with a “v” after the labels if another fluorescence was observed in the cytoplasm. Bars = 20 µm.

**Figure 5 metabolites-11-00571-f005:**
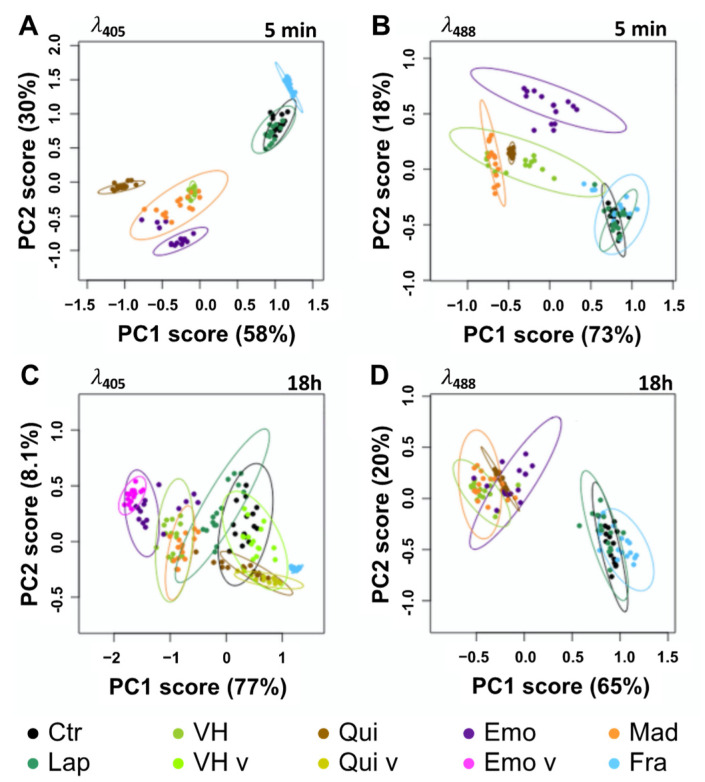
PCA of normalized average spectra obtained by SImaging analysis of BY-2 cells treated or not (Ctr) with 25 µM of phenolic compounds. (**A**,**B**) Differences observed after 5 min treatments and (**C**,**D**) 18 h with vismione H VH, madagascine Mad, emodin Emo, quinizarin Qui, fraxetin Fra, and lapachol Lap. Spectra observed at λ_405_ (**A**,**C**) and λ_488_ (**B**,**D**) in control and treated cells. Spectra found in the primary vacuole at λ_405_ after 18 h are specified with a “v” after the labels if another fluorescence was observed in the cytoplasm. Ellipses are representative of qualitative differences with a *p* ≤ 0.05 for the PCA analysis of normalized average spectra.

**Figure 6 metabolites-11-00571-f006:**
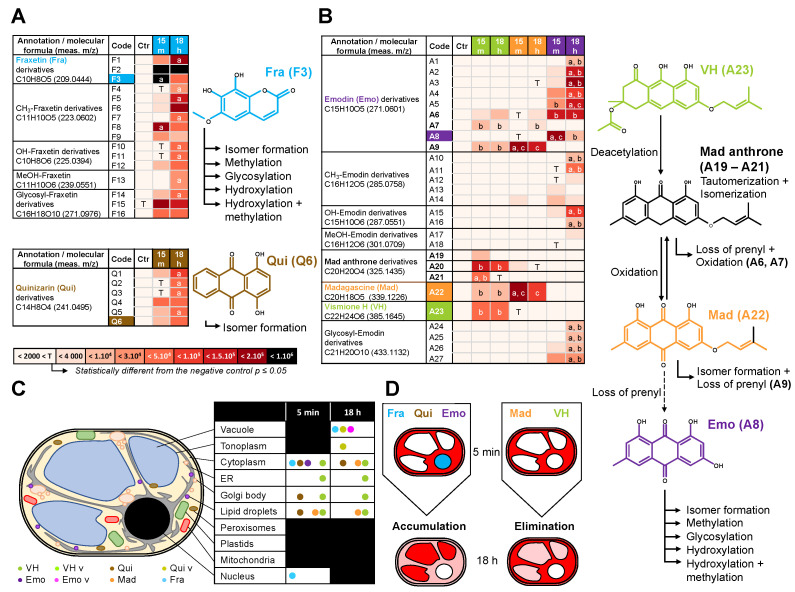
Hypothetical metabolization reactions according to non-targeted analysis of treated BY-2 cells. *m/z* of parent and daughter ions from references and related metabolites detected by UPLC-HRMS/MS analysis of methanolic extracts from BY-2 cells treated for 15 min (15 m) and 18 h with (**A**) 50 µM of fraxetin Fra (blue), (**B**) quinizarin Qui (brown), or (**C**) vismione H VH (green), emodin Emo (purple) and madagascine Mad (orange). meas. *m/z* = measured mass to charge ratio of the most intense adduct detected. Proposed reactions such as hydroxylation (+OH), methylation (+CH_3_), hydroxylation/methylation (+MeOH), glycosylation (+Sugar) and loss of prenyl which may occur in the metabolization processes. The arrows indicate hypothetical reactions according to the references (colored label) and 54 annotations absent or with an area below the significant threshold in negative control samples (<4000). Statistical analysis was performed on area from annotated metabolites (*n* = 3) using Levene with Kruskal–Wallis tests followed by a Dunnett’s post-hoc test. Significant differences (*p* ≤ 0.05) between area of metabolites from (a) BY-2 cells treated 15 min and 18 h or (b ≠ c) between treatments with anthranoids. (**D**) Conjuncture of the fluorescence localization in BY-2 subcellular compartments observed by SImaging and the non-targeted metabolomic analysis proposing that Fra, Qui, and Emo derivatives accumulate into the vacuole, while not prenylated anthranoids VH and Mad being mostly metabolized into other sub-cellular compartments.

**Table 1 metabolites-11-00571-t001:** Main photophysical characteristics of fluorescent species.

Compounds Name	[C] (µM)	λabs (nm)	ε (104 M^−1^ cm^−1^)	λEx (nm)	λEm (nm)	ΦF (%)	SI λEm (nm)
Vismione H (LH_2_) ^a^	2.6	398	1.1	396	481	3.9	503
Vismione H (LB) ^b^		403	1.40	402	532	31.0	530
Vismione H (LNH_4_^+^) ^c^		403	1.93	403	533	24.0	530
Vismione H-Ca (LCa) ^d^		404	1.36	404	549	19.8	548
Vismione H-Mg (LMg) ^e^		407	1.40	404	550	18.6	548
Vismione H-BSA (LBSA) ^f^		404	1.21	404	517	23.5	530
Emodin (LH_3_) ^a^	5.55	443	1.68	442	575	0.7	565
Madagascine (LH_2_) ^a^	2.96	437	0.64	446	544	4.8	565
Quinizarin (LH_2_) ^a^	4.16	480	0.38	479	569	15.1	565
Fraxetin (LB) ^b^	4.8	366	1.05	367	492	1.1	521
Fraxetin (LNH_4_^+^) ^c^		399	0.87	382	481	0.8	521
Fraxetin-BSA (LBSA) ^f^		410	2.04	410	490	8.4	503

The absorption (λ_abs_), excitation (λ_Ex_) and emission (λ_Em_) maxima obtained by spectrofluorimetry and SImaging (SI λ_Em_), the molar extinction coefficient (ε) and quantum yields determined for the pure compounds at different concentration ([C]) in model solutions. EtOH/H_2_O 1:1 *v*/*v*, 0.1 M NaCl with ^a^ 0.01 M HCl at pH 2, ^b^ 0.01 M Na_2_B_4_O_7_ at pH 10, ^c^ 0.01 M NH_4_HCO_3_ at pH 10, ^d^ 15 mM CaCl_2_, ^e^ 15 mM MgCl_2,_ or ^f^ H_2_O containing 300 µM BSA. The errors on ε and Φ_F_ are estimated to 10%, the errors on λ are estimated to ±1 nm.

## Data Availability

The data presented in this study are openly available in FigShare at doi:10.6084/m9.figshare.16540122.

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
