# Peer review of "Unravelling the Puzzle of Anthranoid Metabolism in Living Plant Cells Using Spectral Imaging Coupled to Mass Spectrometry"

_metabolites, 2021, doi:10.3390/metabo11090571_

Round 1

Reviewer 1 Report

In general, the manuscript was well written and discussed by the Authors. It was enriched with legible and exhaustively described Figures. Only, the descriptions in the "Materials and Methods" section require a slight modification: the NF and NR coefficients included in the equation used to calculate the quantum yields of the fluorescent species require correlation with their description below (nR and nS); fresh stock solution of VH (Ca2+ chelating assay) contained NaCl - How much volume of NaCl was added to the stock solution?; list phenolic compounds used in the subsection "Plant material and treatment". Descriptions under table 1 also require correction  (“excitation (?Em) and emission (?Em) maxima”).

Author Response

Dear Reviewer 1, 

We would like to thank you for your support and the review report adorned with comments and suggestions to improve our manuscript. The revised manuscript has been modified following your comments for the "Materials and Methods" section.

Reviewer 2 Report

Dear Authors, the complex study you present in this manuscript is quite interesting and well-presented. However, I believe that some minor alterations are required, especially regarding grammar, phrase construction, spelling and punctuation (i.e.: row 27, 38, 39, 40 "emo", 49-50, 91, 541-542, 556 - use abbrviation, 607, etc).

Author Response

Dear Reviewer 2, 

We recognize that this study is quite complex and we would like to thank you for your review report with comments and suggestions regarding grammar or phrase construction. The minor alterations concerning grammar, phrase construction and spelling have been taken into account in the revised manuscript.

Reviewer 3 Report

The manuscript entitled “Unravelling the Puzzle of Anthranoids Metabolism in Living Plant Cells Using Spectral Imaging Coupled to Mass Spectrometry” describes a new strategy based on spectral imaging, confocal microscopy, and non-targeted metabolomics using mass spectrometry to locate and discriminate the fluorescence VH in living plants. The manuscript also brings some interesting information regarding the VH metabolites in plant cells and it collaborates to the elucidation of the metabolism of anthranoids in plants. Moreover, the Figures and Tables are insightful and support the author's conclusions. The manuscript is of interest to the reader of Metabolites, therefore, I recommend its publication.

Author Response

Dear Reviewer 3,

Thank you for your review report and the recommendation for publication. We hope this manuscript will feed many of Metabolites readers to serve as a recurrent basis for studying metabolism/biological activity of anthranoids and other natural fluorescent metabolites in living organisms.

Reviewer 4 Report

This manuscript reports on the metabolism and compartmentation of anthranoids in living cells using tobacco cells. The Authors apply an original approach based on a combination of spectral imaging (SImaging), confocal microscopy and non-targeted metabolomics using UPLC-MS/MS. The methods applied are up-to-date and are used with knowledge and imagination. The conclusions are supported by the results, which can be further developed and used in fluorescence localization/structure/activity relationships studies of anthranoids and other fluorescent compounds of pharmacological interest.

Author Response

Dear Reviewer 4,

Thank you for your review report and also for highlighting the interest of our research in fluorescence localization/structure/activity relationship studies. We hope that this manuscript will indeed provide not only tools for studying the biosynthesis and metabolism of specialized metabolites, but also other points of view to understand relation between the metabolism and biological activities of putative drugs in living organisms.